# Ameliorating Effects of Vitamin K2 on Dextran Sulfate Sodium-Induced Ulcerative Colitis in Mice

**DOI:** 10.3390/ijms24032986

**Published:** 2023-02-03

**Authors:** Shouna Hu, Yan Ma, Ke Xiong, Yanrong Wang, Yajun Liu, Yongye Sun, Yuexin Yang, Aiguo Ma

**Affiliations:** 1Department of Nutrition and Food Hygiene, School of Public Health, Qingdao University, Qingdao 266021, China; 2Institute of Nutrition and Health, School of Public Health, Qingdao University, Qingdao 266021, China; 3National Institute of Nutrition for Health, Beijing 100051, China

**Keywords:** vitamin K2, gut microbiota, inflammatory cytokine, ulcerative colitis

## Abstract

Ulcerative colitis (UC) is a chronic recurrent inflammatory illness of the gastrointestinal system. The purpose of this study was to explore the alleviating effect of vitamin K2 (VK2) on UC, as well as its mechanism. C57BL/6J mice were given 3% DSS for seven days to establish UC, and they then received VK2 (15, 30, or 60 mg/kg·bw) and 5-aminosalicylic acid (100 mg/kg·bw) for two weeks. We recorded the clinical signs, body weights, colon lengths, and histological changes during the experiment. We detected the inflammatory factor expressions using enzyme-linked immunosorbent assay (ELISA) kits, and we detected the tight junction proteins using Western blotting. We analyzed the intestinal microbiota alterations and short-chain fatty acids (SCFAs) using 16S rRNA sequencing and targeted metabolomics. According to the results, VK2 restored the colon lengths, improved the colonic histopathology, reduced the levels of proinflammatory cytokines (such as IL-1β, TNF-α, and IL-6), and boosted the level of the immunosuppressive cytokine IL-10 in the colon tissues of the colitis mice. Moreover, VK2 promoted the expression of mucin and tight junction proteins (such as occludin and zonula occludens-1) in order to preserve the intestinal mucosal barrier function and prevent UC in mice. Additionally, after the VK2 intervention, the SCFAs and SCFA-producing genera, such as *Eubacterium_ruminantium_group* and *Faecalibaculum*, were elevated in the colon. In conclusion, VK2 alleviated the DSS-induced colitis in the mice, perhaps by boosting the dominant intestinal microflora, such as *Faecalibaculum*, by reducing intestinal microflora dysbiosis, and by modulating the expression of SCFAs, inflammatory factors, and intestinal barrier proteins.

## 1. Introduction

Inflammatory bowel disease (IBD) is an idiopathic inflammatory illness of the intestine that clinically comprises ulcerative colitis (UC) and Crohn’s disease (CD). Ulcerative colitis frequently results from rectal mucosa inflammation that spreads from the distal to the proximal side. Due to rapid lifestyle and environmental changes, ulcerative colitis is becoming an increasingly serious problem for global public health [1,2]. The major clinical symptoms of UC include chronic diarrhea, abdominal pain, blood in the stool, and rapid weight loss [3]. The pathogenesis of UC is not clear; however, diet, genetics, immunity, and intestinal microecology are closely related to its development [4,5,6]. Researchers have revealed that UC may be worsened by unfavorable connections between the intestinal inflammatory response and intestinal bacteria, in addition to an imbalance of proinflammatory and anti-inflammatory factors that promotes the development of UC [7,8]. Glucocorticoids and other anti-inflammatory drugs are currently the mainstays of UC treatment [9]. However, the current pharmacological therapies have drawbacks, such as side effects and expensive pricing. Thus, the creation of safe and effective new pharmaceuticals is currently a pressing issue.

Vitamin K deficiency is commonly seen in people with inflammatory bowel disease [10,11]. Vitamin K, one of the fat-soluble vitamins, is essential for human metabolism and health, and it belongs to the class of naphthoquinone group derivatives with the functional activity of chlorophyll quinone [12]. Vitamin K2 (VK2) belongs to a class of fat-soluble vitamins with coagulation functions, and it belongs to the vitamin K family. Researchers have recently reported the role of vitamin K2 in the inhibition of inflammation, antioxidants, and apoptosis induction in cancer cells, and in the prevention of vascular calcification and osteoporosis [13,14,15,16,17]. A recent study demonstrated that feeding mice a diet lacking vitamin K made ulcerative colitis symptoms worse, and according to the results, vitamin K could suppress the immune system by limiting the synthesis of interleukin-6 (IL-6) [18]. Although VK2 deficiency is an important component of UC, its effects on the gut flora within the context of UC are still unknown. In this study, we focused on how VK2 improves UC in mice by controlling the intestinal flora, repairing the intestinal mucosal barrier, and reducing inflammation.

## 2. Results

### 2.1. Vitamin K2 Administration Ameliorated Ulcerative Colitis

As show in Figure 1a, the body weights of the mice in the DSS group remarkably decreased compared with the body weights of those in the CON group, but mice with VK2 treatment restored their body weights (*p* < 0.01). Moreover, according to the evaluation of the DAI index of the mice, VK2 and 5-ASA also effectively alleviated diarrhea and hematochezia (*p* < 0.05, Figure 1b,c). In addition, the VK2 treatment increased the colon lengths and decreased the spleen weights of the mice in the MVK2 and HVK2 groups, compared with the animals in the DSS group (*p* < 0.05, Figure 1d,e). In comparison to the DSS group, the 5-ASA treatment also remarkably increased the colon lengths (*p* < 0.01, Figure 1d).

Hematoxylin and eosin (H&E) staining was used to systematically assess the damage to the intestinal mucosa. The mice in the VK2 and 5-ASA groups showed less inflammatory cell infiltration, relatively intact colonic architectures, less mucosal damage, and lower histology scores than the mice in the DSS group, which displayed loss of crypts, infiltration of the mononuclear cells, severe mucosal damage, and higher histology scores (Figure 2a,b). Periodic acid-Schiff (PAS) staining results (Figure 2a,c) showed that DSS considerably reduced the number of intestinal goblet cells (*p* < 0.05). The oral administration of VK2 and 5-ASA restored the loss of intestinal goblet cells brought on by DSS (*p* < 0.05). According to these results, the VK2 treatment substantially ameliorated the DSS-induced ulcerative colitis.

### 2.2. Intestinal Barrier Protein and Cytokine Expressions

Compared with those of the mice in the CON group, the IL-6 and IL-1β secretion levels in the colon tissues of the mice in the DSS group were considerably higher (*p* < 0.01, Figure 3a,c). TNF-α also increased, but this was not statistically significant (*p* > 0.05, Figure 3b). Compared with those of the DSS group, the colonic TNF-α and IL-1β levels of the mice treated with various dosages of VK2 were all reduced to varying degrees, and especially in the MVK2 group (*p* < 0.01, Figure 3a,b). The mice supplemented with VK2 presented increased anti-inflammatory cytokine expression (IL-10) (*p* < 0.05, Figure 3d). Moreover, compared to the DSS group, the 5-ASA supplementation considerably reduced the colonic TNF-α levels (*p* < 0.01, Figure 3b) and increased IL-10 levels (*p* < 0.05, Figure 3d).

Furthermore, in the DSS group, the protein expression levels of ZO-1 and occludin in the colon tissues of the mice were considerably lower than those in the CON group (*p* < 0.01, Figure 3g,h). After supplementation with VK2 and 5-ASA, ZO-1 and occludin protein expression levels were elevated compared with the DSS group, with statistically different results in the MVK2 and 5-ASA groups (*p* < 0.01, Figure 3g,h). In addition, the reduction in the protein expression of muc2 induced by DSS was reversed in the MVK2, HVK2, and 5-ASA groups (*p* < 0.05, Figure 3f).

### 2.3. Gut Microbiota Diversity and Composition

Alpha diversity analysis demonstrated that DSS modeling reduced the diversity of the mouse intestinal flora, as evidenced by a marked decline in the ACE, Chao1, Shannon, and Simpson indices (Figure 4a). The VK2 and 5-ASA interventions did not restore the reduction caused by the DSS. The DSS and CON groups had different intestinal microbial compositions, while the intestinal flora compositions of the VK2 and 5-ASA groups were close and markedly different from those of the DSS group (Figure 4b).

According to a phylum-level structural study of the gut microflora (Figure 4c), DSS modeling dramatically improved the abundances of Verrucomicrobiota, Proteobacteria, and Deferribacteres (*p* < 0.05, Figure 4e,g,h), while it significantly decreased the abundances of Firmicutes and Cyanobacteria, and also the Firmicutes/Bacteroidetes (F/B) ratio (*p* < 0.05, Figure 4d,f,i). The VK2 treatment prominently reversed the change in the Deferribacteres abundance caused by the DSS treatment (*p* < 0.05, Figure 4h), and it also reversed the changes in the Proteobacteria and Cyanobacteria abundances and the F/B ratio in the DSS group to a different extent, but not substantially (Figure 4f,g,i). Among all the groups, the MVK2 group saw the best effect.

The mice receiving 30 mg/kg·bw of vitamin K2 supplementation had the greatest improvements in their ulcerative colitis; thus, we performed a genus-level clustering analysis among the 5-ASA, MVK2, CON, and DSS groups (Figure 5a). According to the results, *Akkermansia*, *Bacteroides*, *Butyricimonas*, *Erysipelatoclostridium*, *Fournierella*, *Mucispirillum*, *Parabacteroides*, *Parasutterella*, *Romboutsia*, *unclassified_Desulfovibrionaceae*, *unclassified_Anaerovoracaceae*, and *uncultured_Bacteroidales-bacterium* were substantially more abundant in the DSS model group compared with the CON group. Moreover, *Anaerotruncus*, *Butyricoccus*, *Colidextribacter*, *Monoglobus*, *NK4A214_group*, *Oscillibacter*, *Peptococcus*, *Prevotellaceae_NK3B31_group*, *Roseburia*, *Rothia*, *Ruminococcus*, *Tyzzerella*, *[Eubacterium]_xyianophilum_group*, *unclassified_Clostridia*, *unclassified_Oscillospiraceae*, *unclassified_RF39*, and *unclassified_UCG_010* were substantially decreased in the DSS group (*p* < 0.05, Figure 5b). After the 5-ASA intervention, *Akkermansia* was substantially higher in the DSS group, while *Actinomyces*, *Haemophilus*, *Intestinimonas*, *Mucispirillum*, *Neisseria*, *Prevotella_7*, *Romboutsia*, *Rothia*, *Streptococcus*, *Veillonella*, *unclassified_Desulfovibrionaceae*, and *unclassified_Ruminococcaceae* were substantially decreased (*p* < 0.05, Figure 5d). Compared with the DSS group, the VK2 intervention substantially increased the levels of *Akkermnsia*, *Faecalibaculum*, *Muribaculum*, *Peptococcus*, *Streptococcus*, *Eubacterium_ruminantium_group*, *unclassified_RF39*, and *uncultured_UCG_010*, and it decreased the levels of *Butyricimonas*, *Haemophilus*, *Lactibacillius*, *Mucispirillum*, *Parabacteroides*, and *unclassified_Anaerovoracaceae* (*p* < 0.05, Figure 5c). These genera are the crucial bacteria for the role of VK2 in ulcerative colitis.

To explore the effects of the different vitamin K2 doses on the intestinal flora of the UC mice, we performed a genus-level cluster analysis of the bacteria on the groups that received the VK2 intervention (Figure 6a). According to the results, compared with the MVK2 group, *GCA-900066575*, *Gemella*, *Eubacterium_ruminantium_group*, *unclassified_Clostridia*, and *unclassified-RF39* were substantially decreased in the LVK2 group (*p* < 0.05, Figure 6b). In the MVK2 and HVK2 groups, the *Christensenellaceae_R_7_group*, *GCA_900066575*, *Muribaculum*, *Peptococcus*, *Prevotellaceae_UCG_001*, *Rothia*, *Streptococcus*, *Eubacterium_ruminantium_group*, *Eubacterium_siraeum_group*, *unclassified_Clostridia*, *unclassified_Desulfovibrionaceae*, and *unclassified_Lachnospiraceae* were substantially more abundant in the MVK2 group (*p* < 0.05, Figure 6c). We observed changes in the abundances of the crucial bacteria in the LVK2 and HVK2 groups (*p* < 0.05, Figure 6d). Compared with the MVK2 group, harmful bacteria, such as *Butyricimonas*, *Parabacteroides*, *Mucispirillum*, *unclassified_Anaerovoracaceae*, and *unclassified_UCG_010*, were more abundant in the LVK2 and HVK2 groups, while *Faecalibaculum*, *Peptococcus*, and *Eubacterium_ruminantium_group* were less abundant in both groups. According to the LEfSe analysis (LDA > 3.5, Appendix A), *Mucispirillum* and *Parabacteroides* had the highest LDA values in the LVK2 group. The most highly represented bacteria in the HVK2 group were *unclassified_Atopobiaceae* and *Erysipelatoclostricium*.

We performed a Spearman’s correlation analysis on the 30 most abundant genera in the VK2 groups, and we visualized the results as a heatmap (Appendix A). According to Appendix A, there were two competing clusters: one cluster consisted of *Mucispirillum*, *unclassified_Ruminococcaceae*, *Bacteroides*, *Parabacteroides*, *Butyricimonas*, *Parasutterella*, *uncultured_Bacteroidales_bacterium*, *unclassified_Atopobiaceae*, *unclassified_Muribaculaceae*, and *Erysipelatoclostridium*, and the other consisted of *unclassified_Clostridia_UCG_014*, *[Eubacterium]_ruminantium_group*, *Alistipes*, *[Eubacterium]_siraeum_group*, *Prevotellaceae_UCG_001*, *unclassified_Desulfovibrionaceae*, *unclassified_Lachnospiraceae*, and *Lachnospiraceae_NK4A136_group*. *Erysipelatoclostridium*, which is the biomarker of LVK2 and HVK2, had strong negative correlations with beneficial bacteria, such as *[Eubacterium]_siraeum_group* and *unclassified_Clostridia_UCG_014*, and it had positive correlations with harmful bacteria, such as *Mucispirillum*. The *[Eubacterium]_siraeum_group* was the dominant bacterium in the MVK2 group, and *Mucispirillum* had the highest LDA value in the DSS group. 

### 2.4. Production of Targeted Short-Chain Fatty Acids in Cecal Contents

We conducted a quantitative analysis of the SCFAs to determine whether the ameliorative impact of VK2 on UC is connected with the generation of SCFAs. According to the results, the acetic acid, butyric acid, and total acid levels were considerably lower in the DSS group than in the CON group (*p* < 0.01, Figure 7). In addition, the isobutyric acid and valeric acid levels were also lower than those in the CON group, but the differences were not significant (*p* > 0.05, Figure 7). After the VK2 intervention, the acetic acid, propionic acid, and isobutyric acid levels were obviously increased in the MVK2 and HVK2 groups of mice (*p* < 0.05, Figure 7), and they were even higher than those of the mice in the CON group. The 5-ASA group of mice had considerably higher amounts of acetic acid, propionic acid, isobutyric acid, n-valeric acid, and isovaleric acid than the DSS group (*p* < 0.05).

### 2.5. Correlation between Intestinal Flora and Ulcerative Colitis Indexes

The association between the differentially enriched microorganisms and biochemical indicators, or SCFA profiles, was investigated using Spearman’s correlation analysis. According to the correlation analysis, among the DSS, LVK2, MVK2, and HVK2 groups, 22 different bacterial genera were significantly correlated with the barrier protein or inflammatory mediator concentrations in the colon and/or SCFAs in the feces (*p* < 0.05, Figure 8). The bacterium *Mucispirllium*, which was enriched in the DSS group, had strong negative correlations with colon lengths, occludin in the colons, and acetate, butyrate, and total acid levels in the feces; however, it had a substantially positive correlation with the IL-1β in the colon. *Proteus* had strong negative correlations with the propanoic acid, valerianic acid, acetic acid, and total acid levels in the feces, and positive correlations with the colonic muc2 and ZO-1 levels. *Haemophilus* had strong negative correlations with the production of acetate and total acid in feces, and with the colonic muc2, ZO-1, and IL-10 levels, and it had a positive correlation with the colonic IL-1β levels. Furthermore, *Alloprevotella* and *unclassified_Enterobacteriaceae* had substantial connections with the colonic IL-1β levels.

In the intervention groups that received different doses of VK2, the genera *Eubacterium_ruminantium_group, Streptococcus, Eubacterium_siraeum_group, Akkermansia, unclassified_RF39, Gemella, unclassified_Clostridia, Faecalibaculum,* and *uncultured_Clostridiales_bacterium,* which were enriched in the MVK2 group, had strong correlations with the biochemical indicators. *Streptococcus* had positive correlations with the colonic muc2 and ZO-1 levels and negative correlations with the colonic TNF-α and IL-1β levels. The *Eubacterium_ruminantium_group* and *Eubacterium_siraeum_group* had negative correlations with the colonic IL-1β levels. *Gemella* and *Faecalibaculum* had strong positive correlations with the colonic muc2, ZO-1, IL-10, and occludin levels, and negative correlations with the colonic TNF-α and IL-1β levels. *Faecalibaculum* also had positive correlations with the fecal production of propanoic acid, acetic acid, and total acid. The genera *Lactobacillus, Muribaculum, Anaerotruncus, Blautia*, and *TM7_X_* were substantially enriched by the LVK2. According to the correlation analysis, *Lactobacillus* had negative correlations with the colonic occludin and IL-10 levels and colon length, as well as with the fecal production of propanoic acid, acetic acid, and total acid, and it had positive correlations with the colonic TNF-α levels and spleen weight. Among the HVK2-enriched genera, *Leptotrichia* had a negative correlation with the fecal production of isobutyric acid. *Peptostreptococcus* and *Prevotella* had positive correlations with the DAI. *Erysipelatoclostridium* had a positive correlation with the colonic IL-6 levels.

## 3. Discussion

This study examined the effects of VK2 on ulcerative colitis in vivo. We determined the VK2 intervention dose for this experiment based on previous studies [19,20,21] and on the range of recommended VK2 intakes in each country [22,23]. According to the results, VK2 had a protective effect on DSS-induced ulcerative colitis in the mice, as evidenced by increased body weights and colon lengths and decreased DAI scores and spleen weights. The VK2 treatment promoted the relative abundances of beneficial microbes in the mice, and it facilitated the production of SCFAs and the related protein and cytokine expressions. According to the present study, VK2 can prevent and alleviate UC via intestinal flora regulation.

In previous studies, researchers have reported that VK2 deficiency exacerbates the dextran sodium sulfate colitis in mice [18], and that VK2 has significant anti-inflammatory effects in vitro via mechanisms such as the specific inhibition of the nuclear factor kappa B (NF-κB) pathway [13]. However, its anti-inflammatory and antibacterial effects in vivo, and especially in UC, have rarely been reported. Consequently, in this study, we examined the effects of VK2 supplementation on UC in vivo. As expected, the VK2 administration substantially alleviated the ulcerative colitis, and it was mainly by decreasing the DAI index, increasing the expression of anti-inflammatory factor IL-10, muc2, and tight junction proteins (ZO-1, occludin), and by improving the intestinal tissue architecture.

Different from other inflammatory diseases, gut flora dysbiosis plays a significant part in the etiology of ulcerative colitis. According to the results of the alpha diversity measurements based on the OTUs and Chao, Shannon, and Simpson indexes, the VK2 groups had considerably more diverse microbiota than the DSS group, which concurred with earlier reports [24]. The flora diversity of the mice was higher than that of the DSS model group after the VK2 treatment, but not substantially. According to the beta diversity data results, there was substantial clustering separation between the VK2 intervention and DSS groups, which indicated that the VK2 substantially altered the intestinal microbial structure, and that the altered microbiota may be crucial in preventing the development of colitis.

At the phylum level, changes in the Proteobacteria and Deferribacteres abundances were critical in the VK2 treatment of UC. Proteobacteria and Deferribacteres were substantially more abundant in the DSS group. Researchers have reported that Proteobacteria are increased in the flora of adults and adolescents with celiac disease [25]. In the DSS group, Firmicutes and the Firmicutes/Bacteroidota ratio were considerably reduced. A normal F/B ratio is thought to be a sign of normal intestinal homeostasis, whereas an abnormal F/B ratio is a biological condition that is associated with IBD [26].

All these changes were restored to varying degrees after the VK2 intervention, and especially after the intervention with 30 mg/kg·bw of VK2, when the Proteobacteria and Deferribacteres abundances were observably lower than in the DSS group, which was consistent with the findings of experiments with UC-related drugs [27]. The abundance of Verrucomicrobia was considerably higher in the DSS group than in the CON group, and according to the analysis of the genus-level bacteria, *Akkermansia* dominated the changes in Verrucomicrobia in the present study. Recently, researchers have revealed that variations in the *Akkermansia* abundance are related to inflammatory bowel disease; however, the findings have been inconsistent [28,29]. Vigsnaes et al. [30] discovered a considerable reduction in the *Akkermansia* abundance in individuals with inflammatory bowel disease; however, Håkansson et al. [31] found increases in the *Akkermansia* abundances in DSS-treated C57BL/6 and STAT1−/− mice. According to the current study, the changes in *Akkermansia* may be due to two reasons. First, researchers have demonstrated that an energy-dense diet may decrease the *Akkermansia* abundance in both humans and mice [32,33]. In the present experiment, to ensure consistency between the groups, we gavaged the control mice with corn oil, which may have led to changes in the intestinal microenvironment due to the excess nutrients in the intestine. Second, variations in the abundance of bacteria that break down mucus (*Akkermansia*) can modify the intestinal mucus “degradation–production” balance, and the occurrence of colitis leads to enhanced mucus degradation and *Akkermansia* overexpression, which further damages the intestinal mucosal barrier [34]. Numerous endotoxins and pathogenic bacteria enter the circulatory system through intestinal mucosa damage, increasing intestinal permeability and escalating intestinal inflammation and immunological response [35].

At the genus level, the changes in *Butyricimonas* abundance play a crucial role in UC induction and treatment. Contrary to earlier results, the *Butyricimonas* abundance dramatically rose in the model group [29]. In a previous study, *Butyricimonas*, which belongs to the Bacteroidetes species, thrived at high pH levels, but was not tolerant to low ones [36,37]. The substantially decreased SCFA levels in the DSS group resulted in higher PH values, which may explain the increase in the *Butyricimonas* abundance. Following the VK2 treatment, there was a considerable increase in the SCFA concentrations and quantity of SCFA-producing species, and particularly the *Eubacterium_ruminantium_group* and *Faecalibaculum* genera. This change might induce a boost in the abundance of *Butyricimonas*. At the genus level, the *Anaerotruncus*, *Bacteroides*, *Colidextribacter*, *Peptococcus Erysipelatoclostridium*, *Mucispirillum*, *Parabacteroides*, *Parasutterella*, and *Romboutsia* abundances were also important to the prevalence and treatment of the UC, which was consistent with the results of previous studies in which the authors revealed substantial increases in *Bacteroides*, *Romboutsia* [38], *Mucispirillum*, *Parabacteroides* [29], *Erysipelatoclostridium* [39], *Peptococcus* [40], and *Parasutterella* [41], as well as substantial decreases in *Anaerotruncus*, *Colidextribacter* [42], *Ruminococcus*, and *Monglobus* [43] in UC mice, which might have been induced by an imbalance in the Th17/Treg transition, which is crucial for the maintenance of gut immunological homeostasis. Researchers found that *Colidextribacter*, which belongs to *Clostridiales cluster IV* and *Clostridium cluster XIVa* in the phylum Firmicutes, and which produces SCFAs, had a substantial effect on the development and function of Treg cells [42]. In other relevant studies, researchers have demonstrated that *Ruminococcus* and *Monglobus* are positively correlated with the CD4 + T cell subsets and can alleviate inflammation by affecting the Treg cells [43], and that changes in their abundances may lead to an imbalance in the Th17/Treg ratio. According to one report, DSS-induced colitis is influenced heavily by gut microbiota dysbiosis connected with the immunological response [44]. In the current study, the VK2 intervention did not have a dose-dependent ameliorative impact on the ulcerative colitis. We observed the most noticeable effects in the intervention group that received 30 mg/kg·bw of VK2. We conducted a differential marker analysis for the DSS and MVK2 groups to identify the underlying biomarkers and dominant flora mediated by the VK2 treatment. Following the VK2 intervention, the data indicated reversals of the UC-related genera (*Butyricimonas*, *Mucispirillum*, *Parabacteroides*, *Peptococcus*, etc.). Furthermore, *Faecalibaculum*, *Muribaculum*, *Peptococcus*, *Streptococcus*, and the *Eubacterium_ruminantium_group* were enriched in the MVK2 group. *Faecalibaculum* and the *Eubacterium_ruminantium_group* are a new generation of “possibly helpful microbes” with the ability to create SCFAs. They play a crucial role in energy balance, colonic motility, immunological control, and the inhibition of intestinal inflammation [45], and they can stimulate the development of colon Treg cells and reduce inflammation [46]. Researchers have also discovered a link between a decline in *Faecalibaculum* and the development of rheumatoid arthritis, allergies, asthma, and colorectal cancer [47,48]. *Streptococcus* is an independent VK2-producing bacterium that is engaged in glucose metabolism, and it may be advantageous in obesity-associated disorders [49]. Researchers have demonstrated that a reduction in *Muribaculum* results in inflammation, dyslipidemia, and glucose intolerance [50]. VK2 intervention can drastically alter the compositions of microbial communities by decreasing the abundances of harmful bacteria and increasing the abundances of helpful bacteria, and by promoting the synthesis of SCFAs to control the dysregulation of the intestinal microbiota induced by ulcerative colitis produced by DSS.

To further explore the reasons for the differences in the improvement effects of the different VK2 doses on ulcerative colitis, we compared the abundances of the key VK2 bacteria in the three groups. The abundances of crucial bacteria (*Faecalibaculum*, *Streptococcus*, and *Eubacterium_ruminantium_group*) were dramatically decreased in the LVK2 and HVK2 groups, while the *Mucispirillum* abundance was substantially elevated, which researchers have demonstrated promotes colitis in hosts with severe immunological deficits [51]. Furthermore, according to the LEfSe analysis between the two groups of mice in the VK2 intervention group, the crucial LVK2 and HVK2 bacteria were primarily clustered in Bacteroidetes and Actinobacteria, whereas the bacteria in the MVK2 group were mainly clustered in Firmicutes. According to related research, the primary effect of medication treatment on the dysregulated intestinal flora of UC mice is the restoration of the genus abundance levels in Firmicutes [27,52]. According to the results, the differences in the dominant genus species among the groups may be responsible for the inconsistent improvements in the ulcerative colitis with the different VK2 doses.

The VK2-induced UC improvement might also be associated with the adjustment of the inflammatory response. In some studies, the authors have reported that the gut microbial community and bacterial metabolite compositions have substantial impacts on the IBD outcome via the modulation of the inflammatory response [53]. Researchers have reported that short-chain fatty acids from intestinal microbes have anti-inflammatory and immunosuppressive properties, which help to maintain the balance of the gut immune system [54].

In this study, we found that the VK2 treatment substantially increased the production of SCFAs from the microbial community, and especially acetic acid and propionic acid. Researchers have previously demonstrated that short-chain fatty acids modulate the size and function of the colonic Treg pool and prevent DSS-induced colitis [46]. The dynamic Treg/Th17 balance is crucial for maintaining the homeostasis of the intestinal microflora, which is controlled by the SCFA signals derived from the microbiota. FOX3, a particular transcription factor produced by Treg cells, reduces immunological responses and inflammation by inhibiting other immune system cells (such as Th17 cells) and secreting the anti-inflammatory factor IL-10 [44]. In related studies, researchers have demonstrated that VK2 can affect immune system function and substantially inhibit the activated T lymphocytes in the peripheral mononuclear cells of individuals with atopic dermatitis [55,56]. In addition, VK2 can increase IL-10 production and decrease the production of proinflammatory cytokines (TNF-α, IL-6, IL-1β) by inhibiting NF-κB activation [57], which is consistent with the findings of the current study, in which VK2 substantially increased the IL-10 levels and downregulated the levels of proinflammatory facters (TNF-α, IL-1β).

Intestinal mucus is an ordered network of glycoproteins, the main components of which are high-molecular-weight glycoproteins known as mucins (muc1-muc21). It possesses a host-specific glycan structure that prevents bacterial contact with epithelial cells, resists infection, and regulates the equilibrium between immune response and external stimuli [58]. In related studies, researchers have demonstrated that the defective mucin expression in the colon leads to a higher susceptibility to chronic inflammation [59]. ZO-1 and occludin, as tight junction proteins, are important structures that constitute the intestinal mucosal barrier [60]. Some investigators have suggested that damage to the epithelial barrier and activation of the innate immune reaction are the main causes of the DSS-induced colitis that produces intestinal inflammation [44]. In this study, the colonic proteins ZO-1, occludin, and muc2 were substantially less expressed in the DSS group, which indicates that the DSS impaired the integrity of the colonic membrane. After the VK2 intervention, the ZO-1, occludin, and muc2 protein expression levels were substantially increased in the colonic tissue, which indicated that the VK2 intervention restored the colonic membrane integrity to some extent. The ameliorative effect on intestinal membrane integrity may be crucial in the VK2 treatment of UC. In the current investigation, the 30 mg/kg VK2 intervention was the most effective at alleviating the ulcerative colitis injury. However, there were no dose–response effects between the improvement in the related indexes and the VK2 intervention doses, which requires further investigation.

## 4. Materials and Methods

### 4.1. Animals

We purchased 48 male C57BL/6J mice (7 weeks of age) from the Beijing Vital River Laboratory Animal Technology Co., Ltd. (Beijing, China; SCXK Jing 2021-0006), and we maintained them in the animal laboratory of the Institute of Nutrition and Health of Qingdao University. All the mice had unrestricted access to food and water under a specific-pathogen-free (SPF) environment with a 12-h diurnal cycle, a relatively constant temperature of 21 ± 2 °C, and a relatively constant humidity of 45% ± 10%. We performed all the animal experiments according to the guidelines of the Experimental Animal Care and Ethics Committee of Qingdao University (No. 20220311C574820220401105).

### 4.2. Treatment of UC with Vitamin K2

As shown in Appendix A, we randomly distributed 48 mice into 6 groups (*n* = 8). The entire process lasted 21 days. We provided the animals with unrestricted access to tap water supplemented with or without 3.0% (*w*/*v*) DSS (molecular weight: 36–50 kDa) (MP Biomedical Corporation, Irvine, CA, USA) from day 0 to day 7, followed by the intragastric administration of solutions with different VK2 doses (the solvent was corn oil and menaquinone-4 (Sigma-Aldrich, St. Louis, MO, USA)). We administered 1.5% of the DSS solution from days 15 to 21 to maintain the ulcerative colitis model. The interventions for each group were as follows: (1) the control group (CON): tap water for 7 days, followed by the intragastric administration of corn oil for 14 days; (2) the model control group (dextran sodium sulfate (DSS)): 3.0% DSS for 7 days, followed by the intragastric administration of corn oil for 14 days, and 1.5% DSS solution from days 15 to 21; (3) the DSS + 15 mg/kg·bw VK2 group (LVK2): 3.0% DSS for 7 days, followed by the intragastric administration of 15 mg/kg·bw of VK2 solution (the solvent was corn oil) for 14 days, and 1.5% DSS solution from days 15 to 21; (4) the DSS + 30 mg/kg·bw VK2 group (MVK2): 3.0% DSS for 7 days, followed by the intragastric administration of 30 mg/kg·bw of VK2 solution (the solvent was corn oil) for 14 days, and 1.5% DSS solution from days 15 to 21; (5) the DSS + 60 mg/kg·bw VK2 group (HVK2): 3.0% DSS for 7 days, followed by the intragastric administration of 60 mg/kg·bw VK2 solution (the solvent was corn oil) for 14 days, and 1.5% DSS solution from days 15 to 21; and (6) the DSS + 100 mg/kg·bw 5-aminosalicylic acid (5-ASA) group: 3.0% DSS for 7 days, followed by the intragastric administration of 100 mg/kg·bw 5-ASA solution (the solvent was physiological saline) for 14 days, and 1.5% DSS solution from days 15 to 21.

### 4.3. Determination of Colon Histology

We fixed the distal colon tissues with 4% paraformaldehyde for 36 h before embedding them in paraffin. We cut the embedded tissues into 5 μm thick slices. Hematoxylin and eosin (H&E) and periodic acid-Schiff (PAS) staining were used on the sections for histomorphometry and goblet cells observation, respectively. We randomly selected six fields from each sample and imaged them. We used the histologic scoring criteria of Dohi et al. [61] to assess the degree of mucosal injury. The slices were examined under a microscope (Olympus Corporation, Tokyo, Japan) to differentiate the morphological structures of normal and pathological colonic tissue.

### 4.4. Sample Collection

Throughout the experiment, we examined all the mice daily for signs of sickness, and we graded them for pathological characteristics, including fecal status, occult blood, and body weight loss. Based on the grading system displayed in Appendix A [28], we merged the individual scores to create a disease activity index (DAI). After all the treatments, we sacrificed the mice after the administration of general anesthesia. We performed a postmortem examination to assess the external surfaces, thoracic and abdominal cavities, and contents. Before subjecting the samples to 16S rRNA sequencing analysis, we collected them for examination of the gut microflora in the cecal contents, and we maintained them at −80 °C. We used the distal colon for the study of the proinflammatory cytokines (instantly frozen) and histology (fixed in 4% paraformaldehyde).

### 4.5. Determination of Biochemical Indices in Colon

The colonic samples (30 ± 5 mg) were added to nine volumes of 1 × PBS cocktails before homogenizing them with a homogenizer. The supernatant from the centrifugation of the homogenate suspension at 12,000× *g* for 15 min at 4 °C served as the tissue extract. We tested the colon tissues using an ELISA kit to measure the IL-6, IL-1β, IL-10, and TNF-α levels (Wuhan ABclonal Biological Technology Co, Wuhan, China).

### 4.6. Determination of Short-Chain Fatty Acids

We took the feces of eight mice in each group to determine the short-chain fatty acids (SCFAs). We extracted the metabolites in the colon contents via ultrasonic extraction with 50% sulfuric acid in an ice water bath. We quantitatively determined seven kinds of SCFAs by gas chromatography–mass spectrometry. We performed the determination on a DB-WAX (30 m × 0.25 mm × 0.25 µm) (Agilent Technologies, Santa Clara, CA, USA) column with an injection volume of 1 µL and a flow rate of 1 mL/min. We quantified the SCFAs using an internal standard method, and we drew and quantified the standard curves using GCMS Solutions software (Version 2.50, Shimadzu, Kyoto, Japan).

### 4.7. 16S rRNA Gene Sequencing

We randomly selected 48 fecal samples from six groups (*n* = 8/group). To extract the bacterial genomic DNA, we used the Power Soil DNA isolation kit (Mobio, Carlsbad, CA, USA). We combined the adapter sequence and barcode sequence using the common primer pairs (forward primer: 5′-ACTCCTACGGGAGGCAGCA-3′; reverse primer: 5′-GGACTACHVGGGTWTCTAAT-3′) to amplify the V3–V4 region of the bacterial 16S rRNA gene. We sequenced the V3–V4 region of the 16S rDNA from the intestinal microbiota, and we examined it using high-throughput sequencing technology from Solexa (Version GA II, Illumina, San Diego, CA, USA).

We combined the paired-end reads using the sequencing platform Illumina Novasep 6000 FLASH software (Version 1.2.7). We quality-filtered the splicing sequences using Trimmomatic software (Version 0.33), and we eliminated the chimeras using UCHIME software (Version 8.1) to produce the high-quality target sequences. We used UCLUST to pick open the reference operational taxonomic units (OTUs) at a 97% sequence identity. We then aligned the representative sequences of each OTU using PyNAST, and we assigned them based on the SILVA138 database.

We used QIIME2 (https://qiime2.org/, accessed on 16 May 2022) to examine the alpha diversity. The ACE, Chao, Shannon, and Simpson indices are the alpha diversity indexes that reflect the richness and diversity of the gut microbiota. We used principal coordinate analysis (PCoA) with Bray–Curtis distance matrices to reveal the beta diversity (R Version 2.15.3). We used the linear discriminant analysis (LDA) effect size (LEfSe) to distinguish the distinct bacteria between the groups, with an LDA cutoff of 3.5.

### 4.8. Intestinal Barrier Protein Expression in Colonic Tissue

After washing the colon tissues with ice-cold PBS containing 1 mM phenylmethylsulfonyl fluoride, we collected and lysed them with an extraction solution that included proteinase and phosphatase inhibitors before the homogenization. We centrifuged the tissue lysates at 12,000× *g* for 5 min after incubating them at 4 °C for 20 min, and we collected the supernatant. We used a protein test kit to evaluate the protein contents (Shanghai Epizyme Biomedical Technology Co., Ltd., Shanghai, China). We mixed 24 micrograms of protein with a sodium dodecyl sulfate (SDS) gel loading buffer, and we resolved using 10% sodium dodecyl sulfate–polyacrylamide gel electrophoresis (Wako Pure Chemical Industries, Tokyo, Japan). Subsequently, we transferred the proteins onto a polyvinylidene fluoride membrane (Millipore, Billerica, MA, USA). After blocking the membrane for two hours with a solution of skim milk powder, we incubated the membrane overnight with the primary antibodies for anti-ZO-1 (ab276131 (Abcam, Cambridge, MA, USA)), anti-mucin 2 (Muc2, A4767 (Wuhan ABclonal Biological Technology Co, Wuhan, China)), anti-occludin (ab216327 (Abcam)), and β-actin (AC006 (ABclonal)) at the matching dilutions. We incubated the secondary antibodies (AS014 (ABclonal)) for 1 h at room temperature before we used enhanced chemiluminescent (ECL) detection reagents to identify the protein bands. We performed the quantitative protein expression analysis using ImageJ (Version 1.8.0).

### 4.9. Statistical Analysis

The Student’s *t*-test (unpaired, two-tailed) was utilized to assess the significant thresholds for comparisons between two distinct groups. To determine the statistically significant difference, one-way analysis of variance (ANOVA) with LSD comparison adjustment was used for more than two groups. The GraphPad Prism 8.0 program (GraphPad Software, San Diego, CA, USA) and SPSS version 26.0 for Windows (SPSS, Inc, Chicago, IL, USA) were used to conduct the statistical analyses. The R Programming Language was used to perform Spearman’s correlation between microorganisms and UC-related factors (R version 4.1.2). We presented the findings as means ± SDs, with a significance threshold of *p* < 0.05.

## 5. Conclusions

According to the findings of the current investigation, the VK2-mediated alterations in the microbial community structure play an important role in decreasing ulcerative colitis. The increase in the number of SCFA-producing bacteria and their metabolites induced by VK2 may also help to maintain the microbial equilibrium in the colon and improve the colonic epithelial integrity.

## Figures and Tables

**Figure 1 ijms-24-02986-f001:**
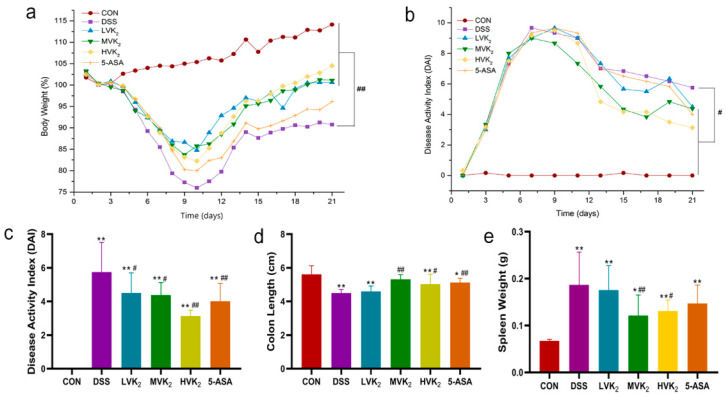
Vitamin K2 (VK2) administration ameliorated ulcerative colitis. (**a**) Daily weight variation. (**b**) Daily disease activity index (DAI) changes. (**c**) DAI scores on day 21. (**d**) Colon lengths. (**e**) Spleen weights. We report the results as means ± SDs, and *n* = 8. We determined the statistical significance using a one-way ANOVA with LSD comparison adjustment. * *p* < 0.05 and ** *p* < 0.01 in comparison to the CON group; # *p* < 0.05 and ## *p* < 0.01 in comparison to the DSS group.

**Figure 2 ijms-24-02986-f002:**
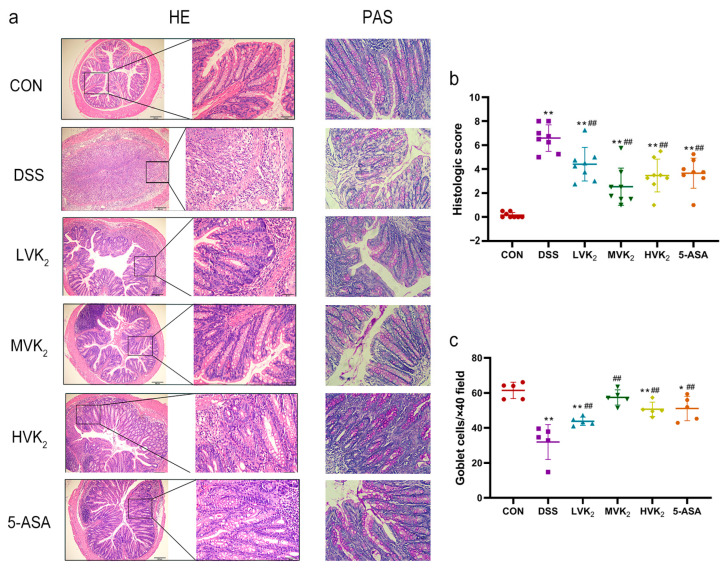
Histological observations of colons. (**a**) Colon H&E and PAS staining in each group. (**b**) Histopathological scores. (**c**) Goblet cell numbers. We report the results as means ± SDs, and *n* = 5–8. We determined the statistical significance using a one-way ANOVA with LSD comparison adjustment. * *p* < 0.05 and ** *p* < 0.01 in comparison to the CON group; ## *p* < 0.01 in comparison to the DSS group.

**Figure 3 ijms-24-02986-f003:**
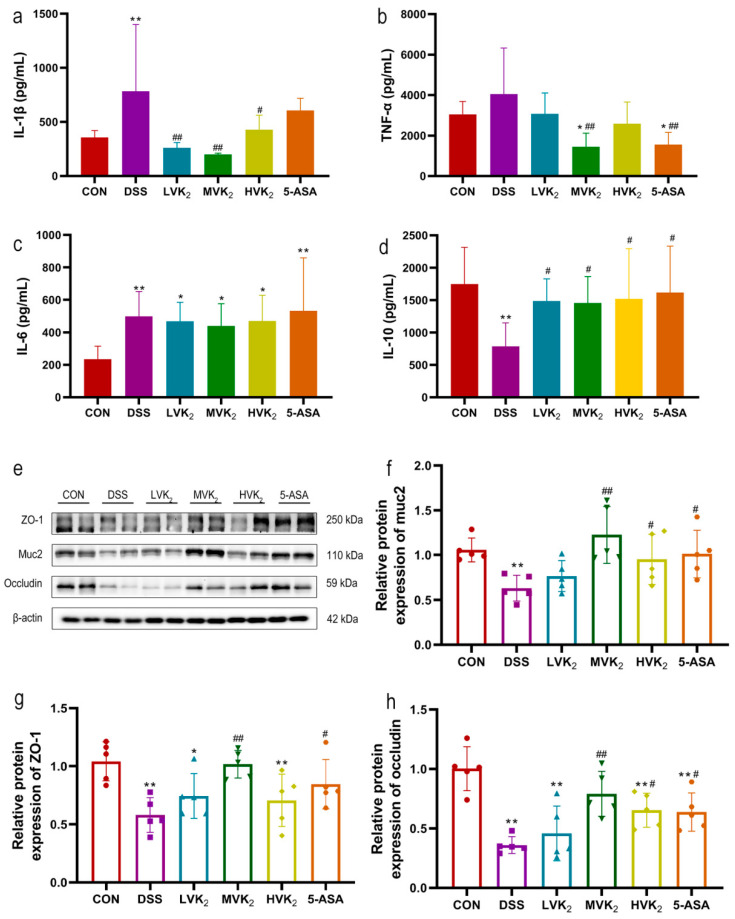
Effects of vitamin K2 on inflammatory cytokine and intestinal tight junction expression. (**a**) IL-1β colonic levels. (**b**) TNF-α colonic levels. (**c**) IL-6 colonic levels. (**d**) IL-10 colonic levels. We report the results as means ± SDs, and *n* = 8. (**e**–**h**) We measured the relative protein expression levels of muc2, ZO-1, and occludin using Western blotting. We report the results as means ± SDs, and *n* = 5. We determined the statistical significance using a one-way ANOVA with LSD comparison adjustment. * *p* < 0.05 and ** *p* < 0.01 in comparison to the CON group; # *p* < 0.05 and ## *p* < 0.01 in comparison to the DSS group.

**Figure 4 ijms-24-02986-f004:**
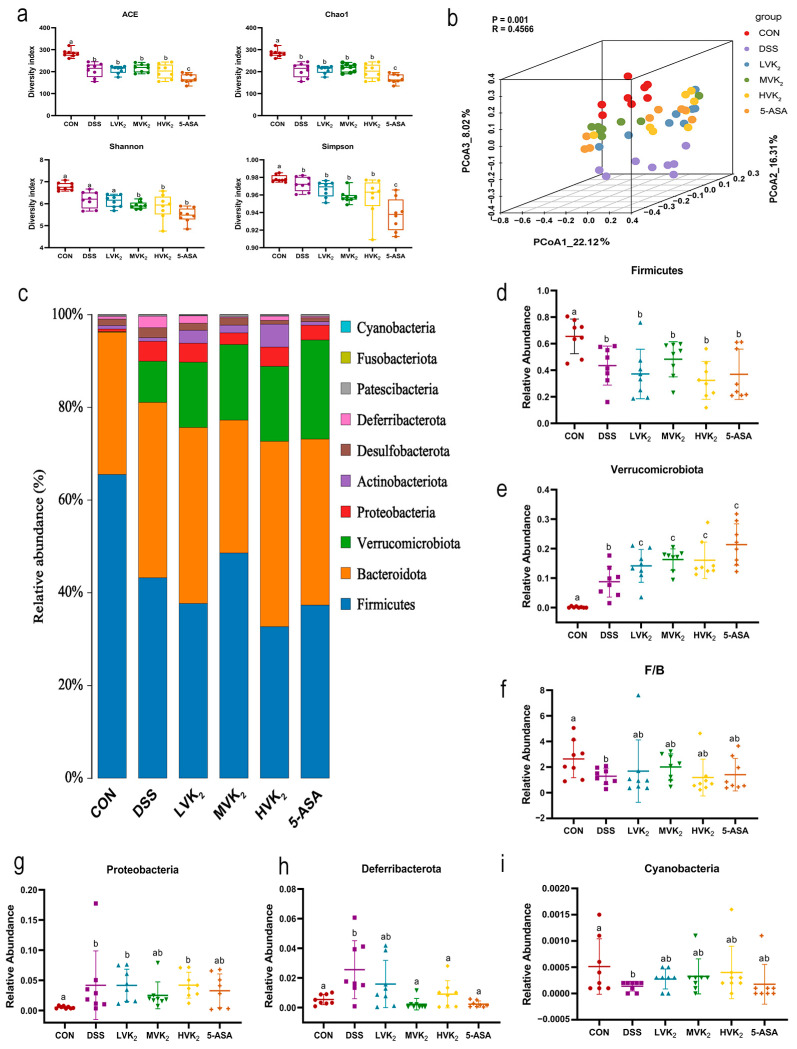
Intestinal flora diversities and compositions in various groups. (**a**) Alpha diversity. (**b**) PCoA analysis (Bray–Curtis distance matrices). (**c**) Bar plots of taxonomic compositions at the phylum level. (**d**–**i**) Different bacteria at the phylum level. We report the results as means ± SDs, and *n* = 8. Based on the one-way ANOVA with LSD multiple comparison method, different letters denote that results were markedly different at the level of *p* < 0.05.

**Figure 5 ijms-24-02986-f005:**
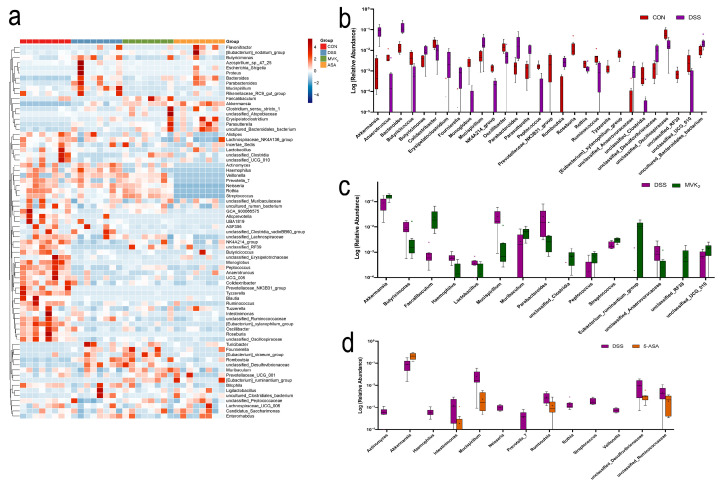
Vitamin K2 (VK2) changed relative abundances of intestinal flora at genus level. (**a**) Community heatmap of relative abundances of intestinal flora at genus level in CON, DSS, MVK2, and 5-ASA groups. Blue indicates less change in a specific bacterium within the four groups, and red indicates more change. (**b**) Different bacteria (*p* < 0.05) between CON and DSS groups. (**c**) Different bacteria (*p* < 0.05) between DSS and MVK2 groups. (**d**) Different bacteria (*p* < 0.05) between DSS and 5-ASA groups. We report the results as means ± SDs, and *n* = 8. We used the Student’s *t*-test (unpaired, two-tailed) to calculate the significance levels.

**Figure 6 ijms-24-02986-f006:**
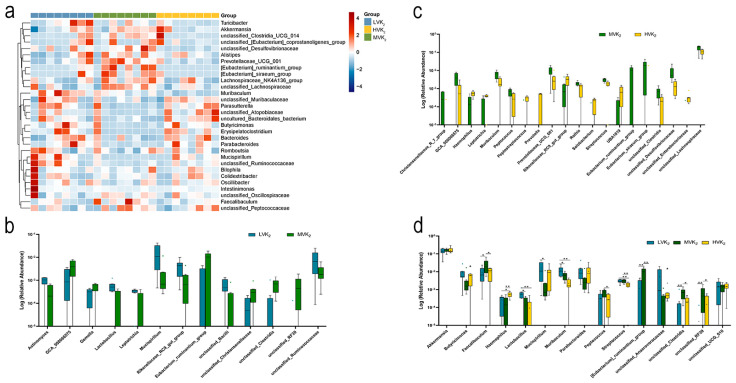
Differences in specific microbial taxa between VK2 intervention groups. (**a**) Community heatmap of relative abundances of gut microbiota at genus level in LVK2, MVK2, and HVK2 groups. Blue indicates less change in a specific bacterium within the four groups, and red indicates more change. (**b**) Different bacteria (*p* < 0.05) between LVK2 and MVK2 groups. (**c**) Different bacteria (*p* < 0.05) between MVK2 and HVK2 groups. We report the results as means ± SDs, and *n* = 8. We used the Student’s *t*-test (unpaired, two-tailed) to calculate the significance levels. (**d**) Comparison of abundance differences in crucial bacteria in vitamin K2 groups. We determined the statistical significance using a one-way ANOVA with LSD comparison adjustment. * *p* < 0.05 and ** *p* < 0.01 in comparison to the MVK2 group.

**Figure 7 ijms-24-02986-f007:**
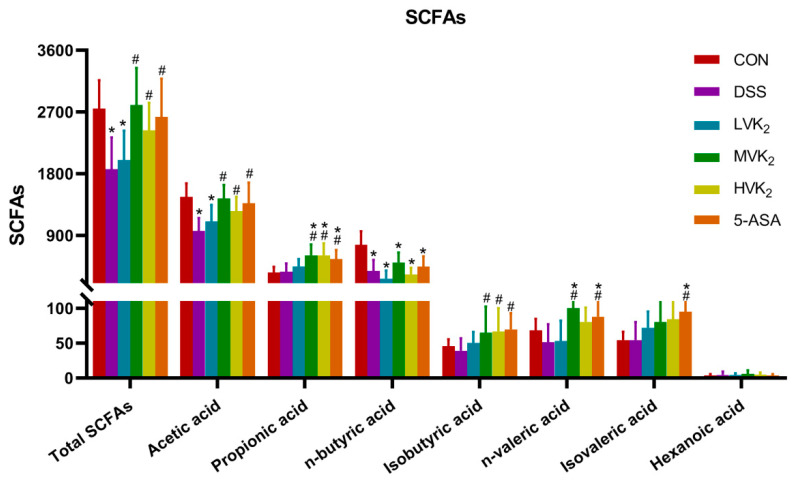
Levels of different SCFAs in cecal contents. We report the results as means ± SDs, and *n* = 8. We determined the statistical significance using a one-way ANOVA with LSD comparison adjustment. * *p* < 0.05 in comparison to the CON group; # *p* < 0.05 in comparison to the DSS group.

**Figure 8 ijms-24-02986-f008:**
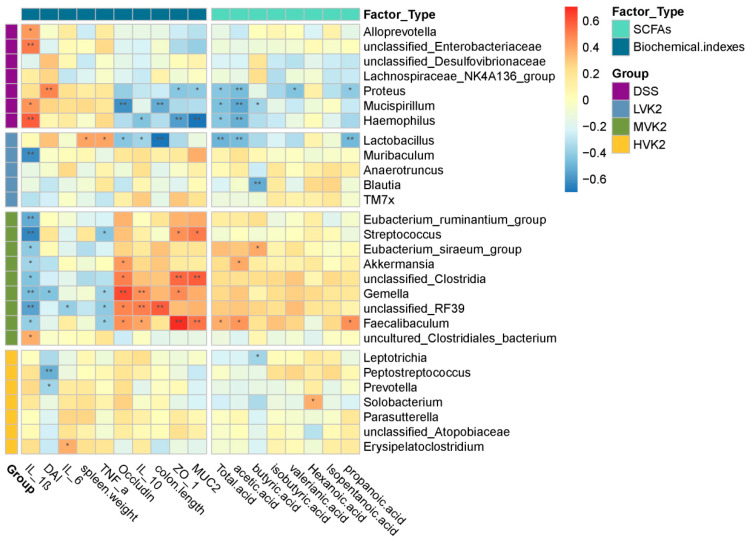
Correlations between intestinal flora and ulcerative colitis indexes. Spearman correlations between intestinal microbiota and biochemical indexes or SCFAs in DSS, LVK2, MVK2, and HVK2 groups. Blue and red colors indicate negative and positive correlations, respectively. The strength of the Spearman correlation correlates directly with the color intensity. * *p* < 0.05 and ** *p* < 0.01.

## Data Availability

No new data were created or analyzed in this study. Data sharing is not applicable to this article.

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
