# Peer review of "Ameliorating Effects of Vitamin K2 on Dextran Sulfate Sodium-Induced Ulcerative Colitis in Mice"

_ijms, 2023, doi:10.3390/ijms24032986_

Round 1

Reviewer 1 Report

The current article entitled “Ameliorating Effects of Vitamin K2 on Dextran Sulfate Sodium-Induced Ulcerative Colitis in Mice” by Hu et al. investigated the vitamin K2 ameliorate the DSS induced colitis in mice. They used three different doses of K2 in there experimental set up and 5 ASA as a positive control. They observed the defects in the colon after DSS with respect to goblet cell number, cytokine profile, tight junction proteins and all were rescued after K2 treatment in these mice. Further they showed altered gut microbiota and short chain fatty acids rescued after K2 treatment. Although this study designed and performed well but fails to investigate the mechanistic aspects of K2 on DSS induced colitis.

Comments:

1.     In figure 1, authors claim about the goblet cells need to be validated by staining colon sections with alcian blue PAS staining and quantifying them.

2.     In figure 1, Defects in colonic sections need to be quantified by using histopathological scoring system in each group.

3.     Does vitamin K2 directly acts on intestinal epithelial cells or it modulates the pro- or anti-inflammatory cytokines secreted by immune cells?

4.     Whether K2 exerts direct effects and modulate the tight junction proteins.

5.     As K2 is known to be produced by intestinal microbiome or the product of the bacterial metabolism such as bacteroids, dose authors observed any differences in that population after DSS colitis in the microbiome data.

Reviewer 2 Report

Shouna Hu et al. describes suppressing effects of vitamin K2 (VK2) on DSS-induced colitis in male mice. They also tried to clarify the mechanistic insights by analyses of bacterial flora and short-chain fatty acid (SCFAs) in the cecal contents, expression of proinflammatory cytokines (IL-6, IL-1β, IL-10, and TNF-α) expression in the colon, and expression of intestinal barrier proteins (ZO-1, Nucin1, occludin) in the colon.

The findings are of interest. However, I would like to points several issues.

1) Abstract should be concisely rewritten according to the aim, experimental procedures, results, and conclusion.

2) The reason for selecting VK2 for a potential compound that has inhibiting effects on experimental colitis is weak. Why serum VK2 was not determined in all groups? 

3) Expression of MUC2 and ZO-1 in the MVK2 is the highest among the VK2-treated groups, suggesting no dose-response effects. Why?

4) Comparison with the 5-ASA group should be carefully described in detail.

5) Legend of Figure S1 is wrong. 

6) The subheadings in the M & M section should be checked.

7) The rationale for dose-selection of VK2 is lacking.

8) “2.3 Determination of Colon Histology” should be rewritten.

9) Cecal content instead of colon content was used for bacterial flora and SCFAs. The reason for this should be given.

10) In this study the proximal colon was used for analysis of pro-inflammatory cytokines’ expression, middle colon for RNA isolation, and distal colon for histopathological examination. In the colitis model of mice treated with DSS, distal colon is affected, and other parts (cecum, proximal and middle colons) are intact. This should be reconsidered.

11) Expression of data: n=3 should not be expressed mean+/-SD. Statistical analysis should be reconsidered.

12) English should be edited. 

Reviewer 3 Report

The authors observed that vitamin K improved inflammation in DSS-induced colitis mouse model. The possible mechanism was that changes in the intestinal microbiota and decreased short-chain fatty acid production reduced the levels of inflammatory cytokines and increased the levels of immunosuppressive cytokines. The study design, analysis methods, and conclusions are acceptable and of interest to the reader. No major corrections are necessary, but there are a few minor errors that need to be corrected, including the following.

P13 2.2. Treatment of UC with Vitamin K2

“(5) DSS + 60mg/kg·bw VK2 group (MVK2)” might be a mistake of “(5) DSS + 60mg/kg·bw VK2 group (HVK2)”

Round 2

Reviewer 2 Report

The revised manuscript has greatly been improved.